# Phylogenetic Characterization of Crimean-Congo Hemorrhagic Fever Virus Detected in African Blue Ticks Feeding on Cattle in a Ugandan Abattoir

**DOI:** 10.3390/microorganisms9020438

**Published:** 2021-02-20

**Authors:** Eddie M. Wampande, Peter Waiswa, David J. Allen, Roger Hewson, Simon D. W. Frost, Samuel C. B. Stubbs

**Affiliations:** 1College of Veterinary Medicine, Animal Resources and Biosecurity, Makerere University, Kampala, Uganda; ewampande@yahoo.co.uk (E.M.W.); pwaiswa4@gmail.com (P.W.); 2Department of Infection Biology, London School of Hygiene and Tropical Medicine, London WC1E 7HT, UK; david.allen@lshtm.ac.uk (D.J.A.); roger.hewson1@lshtm.ac.uk (R.H.); 3Virology and Pathogenesis Group, Public Health England, Porton Down, Salisbury SP4 0JG, UK; 4Department of Infectious Disease Epidemiology, London School of Hygiene and Tropical Medicine, London WC1E 7HT, UK; simon.frost@lshtm.ac.uk; 5Microsoft Research, Redmond, Washington, DC 98052, USA; 6Department of Veterinary Medicine, University of Cambridge, Cambridge CB3 0ES, UK

**Keywords:** Crimean-Congo hemorrhagic fever virus, hemorrhagic fever, viral genomics, *Rhipicephalus (Boophilus) decoloratus*, *Rhipicephalus*, ticks

## Abstract

Crimean-Congo hemorrhagic fever virus (CCHFV) is the most geographically widespread of the tick-borne viruses. However, African strains of CCHFV are poorly represented in sequence databases. In addition, almost all sequence data collected to date have been obtained from cases of human disease, while information regarding the circulation of the virus in tick and animal reservoirs is severely lacking. Here, we characterize the complete coding region of a novel CCHFV strain, detected in African blue ticks (*Rhipicephalus (Boophilus) decoloratus*) feeding on cattle in an abattoir in Kampala, Uganda. These cattle originated from a farm in Mbarara, a major cattle-trading hub for much of Uganda. Phylogenetic analysis indicates that the newly sequenced strain belongs to the African genotype II clade, which predominantly contains the sequences of strains isolated from West Africa in the 1950s, and South Africa in the 1980s. Whilst the viral S (nucleoprotein) and L (RNA polymerase) genome segments shared >90% nucleotide similarity with previously reported genotype II strains, the glycoprotein-coding M segment shared only 80% nucleotide similarity with the next most closely related strains, which were derived from ticks in Western India and Northern China. This genome segment also displayed a large number of non-synonymous mutations previously unreported in the genotype II strains. Characterization of this novel strain adds to our limited understanding of the natural diversity of CCHFV circulating in both ticks and in Africa. Such data can be used to inform the design of vaccines and diagnostics, as well as studies exploring the epidemiology and evolution of the virus for the establishment of future CCHFV control strategies.

## 1. Introduction

Crimean-Congo hemorrhagic fever virus (CCHFV) is a zoonotic pathogen primarily transmitted between vertebrate hosts through the bite of infected ticks, as well as through direct contact with infected livestock, ticks, and bodily fluids [1]. CCHFV is the most geographically widespread of the tick-borne viruses; endemic throughout Africa, Asia, Eastern Europe, and the Middle East, it circulates in an enzootic cycle between animals and ticks, sporadically emerging to cause severe outbreaks of human disease [2]. Outbreak fatality rates of up to 30% have been reported, although CCHF symptoms can range in severity, from mild febrile illness to hemorrhagic fever and multi-organ failure [2].

CCHFV is classified within the genus *Orthonairovirus* (family *Nairoviridae*) and carries a tripartite RNA genome of negative polarity. The three genome segments are labelled small (S), medium (M), and large (L), according to their respective lengths [3]. The ~1.7 kb S segment encodes the viral nucleoprotein (NP) and, in the reverse orientation, an overlapping non-structural (NSs) protein. The M segment is approximately 5.3 kb and encodes a single polyprotein precursor, which is post-translationally cleaved into two envelope glycoproteins (Gn and Gc), several secreted glycoproteins (GP38, GP85, and GP160), and the non-structural M protein, NSm. The pre-Gn precursor also contains a hyper-variable region known as the mucin-like domain (MLD). The ~12.1 kb L segment encodes the viral RNA-dependent RNA polymerase. 

Strains of CCHFV display a high degree of genetic variability for an arthropod-borne virus, with nucleotide diversity as high as 20% for the S segment and 31% for the M segment [4]. Genotypes are assigned to each genomic segment independently based on phylogenetic analysis, with each genotype largely defined by a distinct geographical region: three are African (genotypes I, II, and III), two Asian (genotypes IVa and IVb), and two European (genotypes V and VI). However, limited sampling and uneven representation of CCHFV sequences has hindered phylogeographic and evolutionary analyses [2,5]. This is particularly true for African strains, which are poorly represented in sequence databases in comparison to strains from Europe, Asia, and the Middle East. Furthermore, the majority of CCHFV strains reported to date have been derived from cases of severe human disease, while insufficient sampling from tick vectors and reservoir hosts is almost certainly limiting our knowledge of the natural diversity of CCHFV and the potential for spillover of unanticipated viral variants.

Recent detection of an African genotype III strain following an outbreak in Spain [6], and the presence of African genotype II M segments in viruses obtained from pools of *Hyalomma* spp. ticks in India and China [7,8] has highlighted the need for a more complete genomic dataset in order to better understand the epidemiology of the virus, including viral transmission routes and the extent of re-assortment among genotypes. The potential for such re-assortment is particularly important, as many molecular tests for CCHFV are only capable of detecting a restricted number of closely related genotypes [9]. It also remains to be seen whether recently reported vaccine candidates, including one based on the S and M segments of the Hoti strain (genotype V) [10] and another based on the M segment of the IbAr102000 strain (genotype III) [11], can elicit a protective immune response against viruses belonging to other more distantly related genotypes.

## 2. Materials and Methods

Seventy ticks were collected from cattle, goats, and a pig in an abattoir in Kampala, Uganda between September and December 2019 (Appendix A) as part of a pilot study into the virome of ticks in Uganda. Ticks were immediately frozen in liquid nitrogen following collection and stored at −80 °C until ready for processing. Individual ticks were morphologically identified using taxonomic keys [12] before being crushed using a sterile micropestle and resuspended in 250 µL of phosphate-buffered saline (PBS). Tissue debris was pelleted by low-speed centrifugation (400× *g*) and total nucleic acid (TNA) was extracted from 200 µL of the supernatant using the High Pure Viral Nucleic Acid extraction kit (Roche, Basel, Switzerland) following the manufacturer’s instructions. The extracted nucleic acid was then lyophilized before shipping at ambient temperature to the University of Cambridge, UK for further analysis.

Upon arrival in the UK, TNA was reconstituted in 20 µL of RNase-free water (RFW) supplemented with 2 U of RNaseOUT (Thermo Fisher Scientific, Waltham, MA, USA). Two microliters of reconstituted TNA was subjected to reverse transcription in 20 µL reactions using Superscript III reverse transcriptase (Thermo Fisher Scientific) and primed with 50 ng random hexamers (Thermo Fisher Scientific), as per the enzyme manufacturer’s instructions. Two microliters from each reverse transcription reaction was used as direct input to screen for the presence of nairovirus RNA by (RT-)PCR. PCR was performed in 20 µL reactions, using Phusion DNA polymerase (New England Biolabs, Ipswich, MA, USA) and 0.5 μM pan-nairovirus primers targeting the S segment [13], set up according to the enzyme manufacturer’s instructions. PCR cycling conditions were as follows: incubation at 98 °C for 2 min, followed by 45 cycles of 98 °C for 5 s, 60 °C for 20 s, and 72 °C for 20 s. A final extension was performed at 72 °C for a further 5 min, followed by incubation at 4 °C. The resulting PCR products were visualized by gel electrophoresis, and amplicons of the expected size (~400 bp) were individually purified and confirmed to be CCHFV by Sanger dideoxy sequencing.

Five RT-PCR-positive ticks from Mbarara were selected at random for further characterization by Illumina metagenomic sequencing. Five microliters of TNA from each tick were pooled for a total volume of 25 μL before being treated with 2 U of TURBO DNase (Thermo Fisher Scientific), and incubating for 20 min at 37 °C. DNase-treated RNA was purified using RNA Clean & Concentrator-5 spin columns (Zymo Research, Irvine, CA, USA), eluted in 15 µL of RFW and quantified using the Qubit High Sensitivity RNA assay (Thermo Fisher Scientific ). Illumina sequencing libraries were prepared from 200 ng of purified RNA using the Zymo-Seq RiboFree Total RNA Library kit (Zymo Research), which incorporates a ribosomal RNA depletion step. The resulting library was sequenced on the Illumina NextSeq 500 platform, using the 300-cycle mid-output kit (v2.5, Illumina San Diego, CA, USA). Resulting paired-end reads were imported into CLC Genomics Workbench v7.5.1 (Qiagen, Hilden, Germany) and sequences sharing 95% sequence identity over 35% of their length were assembled de novo. Contiguous sequences corresponding to the CCHFV S, M, and L segments were identified using BLASTn against the redundant nucleotide sequence database with default settings [14].

The CCHFV genome segments (S, M, and L) were independently characterized through phylogenetic analysis. MAFFT v7.427 [15] was used to align the open reading frame of each genome segment against a representative set of CCHFV sequences from GenBank, including members of each genotype (African, Asian, European, etc.) and all available sequences belonging to the genotype to which the Mbarara strain was found to belong (genotype II). For the S and L segments, complete open reading frame (ORF) sequences were analyzed. For the M segment, an 857 bp region at the 5′ end corresponding to the MLD was excluded from phylogenetic analysis due to the extreme variability of the region, which produced sub-optimal alignments. Inclusion of this region did not significantly alter phylogenetic clustering, but bootstrap values were adversely affected. Following trimming of the M segment alignment, the final datasets were 1449, 4323, and 11,838 nt long and included 50, 56, and 46 sequences for S, M, and L segments, respectively. 

Maximum likelihood phylogenies were calculated using IQTree v1.6.11 [16], employing 1000 iterations of the ultrafast bootstrap estimation [17]. Optimal substitution models for each alignment were selected using the ModelFinder algorithm [18] according to Bayesian information criterion scores (GTR+F+I+G4 for S and L segments and GTR+F+R3 for the M segment). Nucleotide identity matrices were generated for each segment using CLUSTAL Omega [19] and intra-genotype nucleotide sequence similarity plots were generated using EMBOSS v6.6.0 [20] Plotcon function across a window size l = 4. Midpoint rooted, maximum likelihood phylogenies and sequence similarity plots were visualized in R v3.6.3 [21] using the ggplot2 [22] and ggtree [23] packages. Genotype II M segment sequences were also analyzed for evidence of recombination using the RDP5 tool [24] and for evidence of episodic diversifying selection using the adaptive branch-site random effects likelihood (aBRSEL) method [25]. Nucleotide alignments of genotype II sequences were also translated into predicted amino acid alignments in Seaview v4.7 [26] and polymorphic and informative sites were identified using the ape v5.0 package [27] in R v3.6.3.

## 3. Results

A total of 70 ticks were collected from cattle (*n* = 48), goats (*n* = 21), and a pig (*n* = 1) (Appendix A). The ticks were morphologically identified as *Rhipicephalus appendiculatus* (*n* = 16), *Rhipicephalus (Boophilus) decoloratus* (*n* = 40), and *Amblyomma variegatum* (*n* = 14). Nairovirus RNA was detected by RT-PCR in 8 of the 70 ticks (11.4%). These 8 ticks were all identified as the African blue tick *R. decoloratus*, and all were picked from cattle. Seven of the CCHFV-positive ticks were sampled from cattle originating from a farm in Mbarara, Western Uganda, and one from a farm in Nakasongola in Central Uganda, as determined based on cattle movement permits. Sanger dideoxy sequencing of the RT-PCR products confirmed the presence of CCHFV and revealed all 8 of the ~400 bp fragments of the S segment to be identical.

Metagenomic sequencing was performed on a pool of five randomly selected (RT-)PCR-positive *R. decoloratus* ticks picked from Mbarara cattle, producing 27,745,476 150 bp paired end reads. A contiguous sequence corresponding to the S segment was 1624 nt in length, with an open reading frame (ORF) of 1449 nt (482 AA), the M segment was 5341 nt (ORF length: 5102 nt; 1700 AA), and the L segment was 11,838 nt (ORF length: 11,838 nt; 3945 AA). The average coverage depth was 6189×, 3882×, and 1833× for S, M, and L, respectively. The full contiguous sequences have been deposited in NCBI GenBank under accession numbers MW452933-MW452935.

Phylogenetic analyses revealed that all three segments of the Mbarara CCHFV strain clustered within the African genotype II (Africa 2) clade, together with strains from the Democratic Republic of Congo, Uganda, and South Africa (Figure 1, Figure 2 and Figure 3). The S and L genome segments were highly similar to previously reported genotype II strains, sharing between 91.79–94.34% nucleotide identity (98.76–99.38% amino acid identity) and 91.57–94.12% nucleotide identity (97.49–98.56% amino acid identity), respectively (Appendix A).

In comparison to the S and L segments, the M segment of the Mbarara strain displayed reduced similarity to genotype II sequences, sharing between 77.54% and 79.62% nucleotide identity (80.45–83.40% amino acid identity). Those sequences with the greatest similarity to the Mbarara strain belonged to several strains from Western India (Gujarat in 2016 and Rajasthan in 2019) and Northern China (Xinjiang in 1979, 2004 and 2016) (Appendix A). As expected, sequence conservation of the M segment was lowest (~40% similarity) in the 5′ region that encompasses the signal peptide and mucin-like domains (Figure 4). However, the variability in genotype II sequences was not limited to the MLD alone; the regions coding for the secreted glycoprotein GP38 and the surface glycoproteins Gn and Gc also displayed a notable degree of intra-genotype nucleotide variability (80–85% similarity). Finally, analysis of the genotype II M segment sequences for recombination and diversifying episodic selection revealed no evidence for either (*p* > 0.05).

Alignment of genotype II predicted amino acid sequences revealed 164 residues unique to the Mbarara strain (singletons) (Table 1). These novel residues were located throughout all three genomic segments, at a frequency ranging from average to above average for the genotype. A majority of singletons (148/164) were observed in the M segment, in which Mbarara strain displayed the greatest number of unique residues across every gene, including the MLD, the non-structural NSm, and the glycoproteins GP38, Gn, and Gc.

## 4. Discussion

Here we describe the detection and characterization of a genotype II strain of CCHFV, present in African blue ticks feeding on cattle from a farm in Mbarara, Uganda. To our knowledge, CCHFV has only rarely been described in this species of tick (*R. decoloratus*): twice in Senegal (1972 and 1975) and twice in Uganda (1978 and 2015) [1,28]. It was not possible to determine whether the virus was derived from within the tick or the host bloodmeal as the specimen was homogenized during the nucleic acid extraction process. However, the tick appears to be the more likely source, as a limited number of experimental studies have found CCHFV viremia to be transient and short-lived in cattle [29]. As a single-host tick species, *R. decoloratus* is unlikely to play a major role in zoonotic transmission of the virus. However, it is possible that these ticks may act as a reservoir for the virus, maintaining infection of the animal host and facilitating the infection of vector species (e.g., *Hyalomma* spp.) through co-feeding [30]. Additionally, several two- and three-host *Rhipicephalus* species have also been shown to support the presence of CCHFV [30]. Therefore, given that *Rhipicephalus* ticks are common throughout Africa, further investigation into their role in CCHFV epizootiology and epidemiology is clearly warranted.

Previous reports of high seroprevalence in livestock and animal handlers throughout Africa (e.g., Bukbuk et al. 2016 [31]), suggest that CCHFV-infected ticks and/or animals can often be found in abattoirs, highlighting the potential for focusing surveillance and control measures around these facilities. Notably, the majority of CCHFV-infected ticks sampled from the abattoir were picked from cattle originating from Mbarara. This district, situated in the cattle corridor of Western Uganda, is a major source of beef and dairy products for the country. Indeed, cattle herds in Nakasongola have been largely sourced from Mbarara, following their depletion during rebel occupation. This may account for the surprisingly high degree of similarity between the Mbarara CCHFV strain and the strain detected in a tick picked from a cow from Nakasongola, and suggests that further sampling of tick CCHFV strains from other regions of Uganda is required in order to better examine transmission dynamics within the country. Possible alternative explanations for the similarity between the CCHFV strains detected in Mbarara and Nakasongola ticks include host-switching of the tick upon arrival at the abattoir, or that the CCHFV-infected tick population originated from the abattoir itself.

Two of the Mbarara strain genomic segments—the nucleocapsid (S) and RNA polymerase (L)—showed a high degree of nucleotide conservation, sharing >90% similarity with other African genotype II sequences, even though these were derived from strains isolated several decades ago. A similar observation was made for viruses from Pakistan and Nigeria [31], suggesting a tendency for these genomic segments to remain genetically stable over time. In contrast, sequence diversity was greatest in the surface-glycoprotein-coding (M) segment, which shared only 78% nucleotide similarity with strains previously reported from Africa. Instead, the M segment sequences bearing the greatest (~80%) nucleotide similarity to the Mbarara strain belonged to two strains sequenced directly from *Hyalomma* spp. ticks in Western India (Gujarat in 2016 and Rajasthan in 2019), as well as several strains from Northern China (Xinjiang). The Chinese strains, one isolated from a rodent (*Euchoreutes naso*) in 1979, and two from *Hyalomma asiaticum* ticks in 2004 and 2016, were first isolated and then passaged in suckling mice prior to sequencing. This should be noted, as the virus may have acquired mutations during adaptation to its mammalian host and therefore may not be representative of the virus where it was initially isolated. Interestingly, the M segments of both the Indian and Chinese strains clustered within the African genotype II clade, however their corresponding S and L segments belonged to the Asian genotype IV, indicating that the M segment had likely been obtained through re-assortment with an African virus. Such a process is made possible through intercontinental movement of CCHFV-infected ticks on livestock and migratory birds [32]. It is unclear whether the relatively low homology of the Mbarara strain M segment to previously reported genotype II strains is due to its novel lineage or simply the result of several decades of evolution. Testing this theory is made difficult by the paucity of closely related sequences. However, the stark dissimilarity to previously reported genotype II M segment sequences, in contrast to the S and L segments, implies recombination or re-assortment with an unidentified viral lineage. Unfortunately, many methods designed to identify instances of recombination and re-assortment are reliant upon the inclusion of the parent lineages in the analysis, which is not possible with the current small sample of genotype II strains.

Amino acid residues unique to the genotype II clade were detected throughout the Mbarara strain genome, although much of the observed diversity, at both nucleotide and amino acid levels, was in the genomic M segment. Unsurprisingly, a high proportion of this variability could be attributed to the region encoding the MLD, a highly glycosylated protein of unproven function but hypothesized to be involved in shielding viral epitopes against the host immune response, based on the role of a similar domain in filoviruses [33]. This hypothesis is supported by a recent epitope-mapping study, which reported a high degree of immune reactivity by CCHFV survivors against the MLD [34]. However, it should be noted that intra-genotype diversity was not exclusively limited to this region of known hyper-variability. A high degree of nucleotide and amino acid variation was also observed in the genes encoding the viral glycoproteins, particularly the secreted glycoprotein GP38. The function of GP38 is similarly unknown, however it is believed to act as a chaperone, assisting in PreGn folding [3]. Interestingly, GP38 has recently shown promise as an antigen following the use of anti-GP38 monoclonal antibodies to protect mice against lethal challenge by CCHFV [35]. Knowledge of the spectrum of GP38 variants in animal and vector reservoirs may therefore prove useful for the design of future vaccines.

## 5. Conclusions

A severely neglected pathogen, the lack of genomic data from African CCHFV makes exploration of the evolutionary history of this novel strain difficult. In addition, a lack of viral data from animal and tick reservoirs has highlighted the need to better characterize the natural variation of CCHFV strains in circulation outside of severe human cases of disease. Indeed, to our knowledge only one other CCHFV strain has been reported from African ticks (IbAr102000). This genotype III virus was isolated in 1966 from a *Hyalomma* tick taken from a camel in Nigeria. Notably, the virus was passaged multiple times over several decades prior to sequencing, likely introducing a number of mutations throughout this period. It is therefore uncertain how representative this sequence is of the original virus, particularly a tick virus adapting to mammalian cell culture. In conclusion, the generation of additional CCHFV genome sequence directly from ticks, such as that presented here, is vital in obtaining a more complete understanding of CCHFV ecology, transmission, and evolution—all of which are necessary for the effective design and deployment of future vaccines and molecular diagnostic tests. 

## Figures and Tables

**Figure 1 microorganisms-09-00438-f001:**
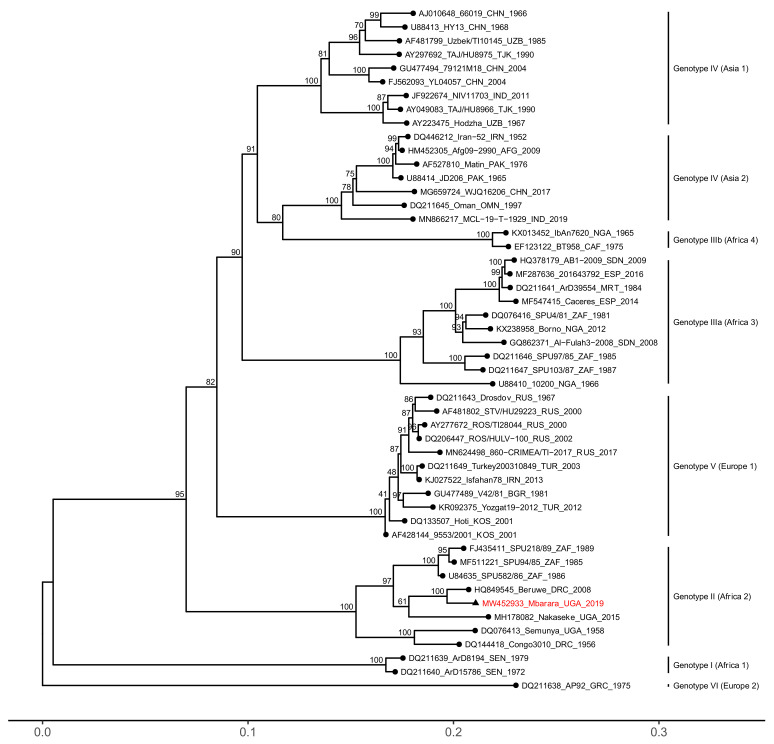
Phylogenetic analysis of the Crimean-Congo hemorrhagic fever virus (CCHFV) S genome segment. Maximum likelihood phylogenies were estimated from complete open reading frame (ORF) nucleotide sequences where available. GenBank accession numbers, strain, country of origin, and year of sampling are listed in taxon names. Numbers at the tree nodes represent bootstrap support values estimated using the ultrafast bootstrap approximation. The Mbarara strain is highlighted in red with a triangular node tip.

**Figure 2 microorganisms-09-00438-f002:**
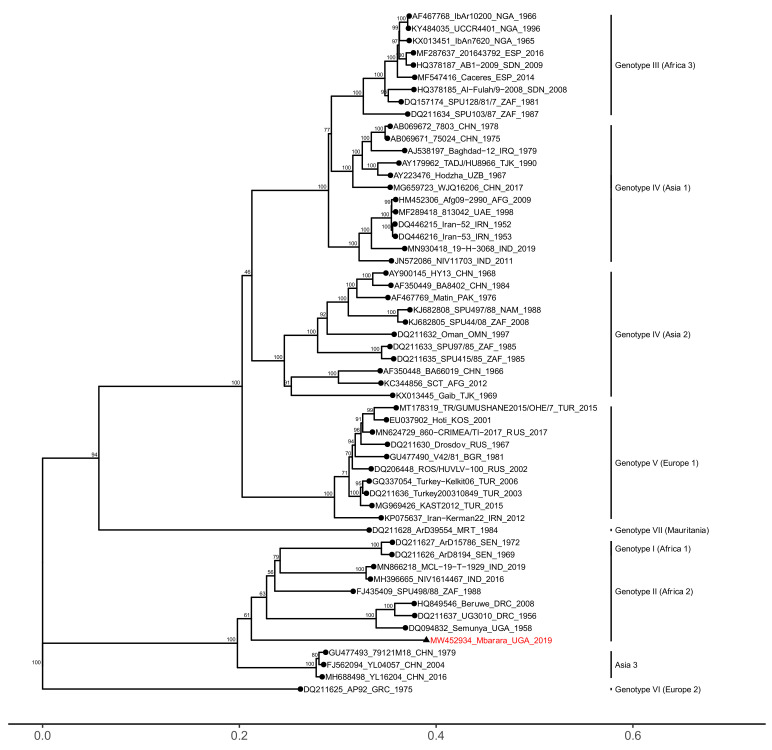
Phylogenetic analysis of the CCHFV M genome segment. Maximum likelihood phylogenies were calculated from nucleotide sequences. The hyper-variable (857 bp) mucin-like domain was trimmed from the 5′ end prior to analysis to facilitate accurate alignment, particularly between genotypes. GenBank accession numbers, strain, country of origin, and year of sampling are listed in taxon names. Numbers at the tree nodes represent bootstrap support values estimated using the ultrafast bootstrap approximation. The Mbarara strain is highlighted in red with a triangular node tip.

**Figure 3 microorganisms-09-00438-f003:**
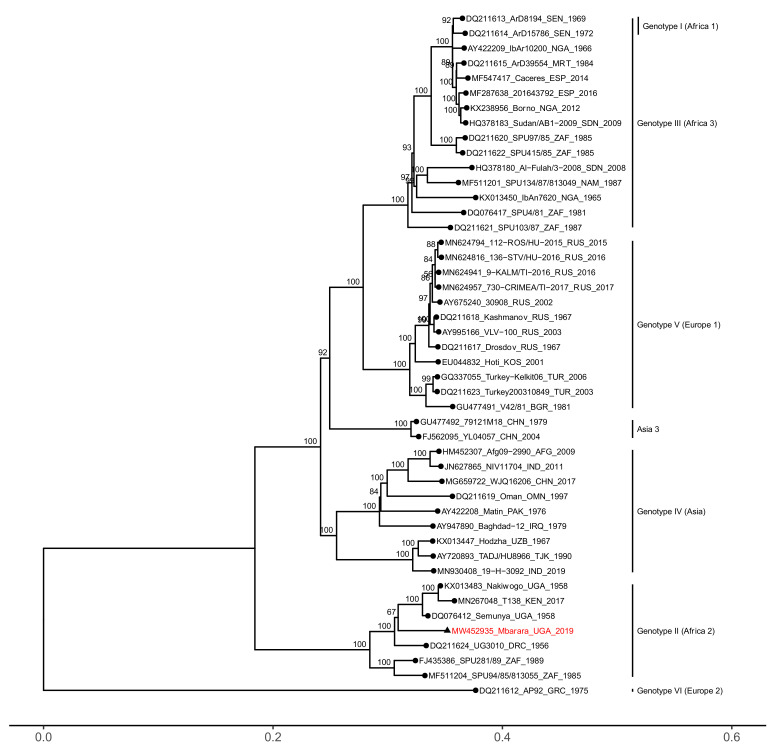
Phylogenetic analysis of the CCHFV L genome segment. Maximum likelihood phylogenies were estimated from complete ORF nucleotide sequences where available. GenBank accession numbers, strain, country of origin, and year of sampling are listed in taxon names. Numbers at the nodes represent bootstrap support values estimated using the ultrafast bootstrap approximation. The Mbarara strain is highlighted in red with a triangular node tip.

**Figure 4 microorganisms-09-00438-f004:**
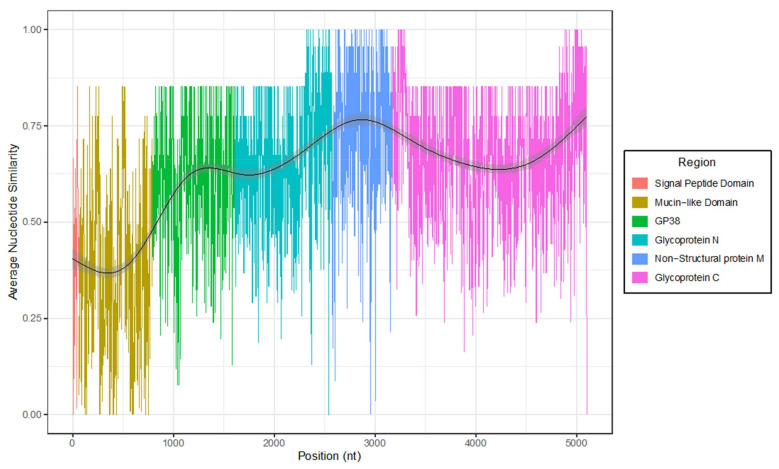
Nucleotide conservation between the glycoprotein precursor (GPC) ORFs of M segments of genotype II strains. Nucleotide sequences of genotype II M ORFs (*n* = 12) were aligned, and the average nucleotide similarity was calculated using a sliding window size of 4. Variability over the context of all mature proteins that are processed from GPC is highlighted.

**Table 1 microorganisms-09-00438-t001:** Predicted amino acid differences in the coding region of the Mbarara strain of CCHFV in comparison to previously sequenced African genotype II strains.

Genome Segment	Gene	Number of Sequences (Full Length)	Gene Length (AA)	Polymorphic Sites * (Informative Sites ^†^)	Median Singleton ^‡^ Count (Range) ^§^	Mbarara Strain Singletons	% Sites Unique to Mbarara Strain
S	NP and Ns	8 (5)	482	11 (3)	1 (0–2)	1	0.2
M	Signal peptide	10 (10)	20	16 (13)	0 (0–1)	5	25.0
	MLD	10 (10)	242	206 (167)	4 (1–29)	76	31.4
	GP38	10 (10)	271	80 (55)	7 (4–14)	22	8.1
	Gn	11 (10)	322	58 (41)	1 (0–4)	12	3.7
	NSm	11 (11)	197	41 (27)	1.5 (0–3)	10	5.1
	Gc	12 (10)	643	103 (53)	2 (0–12)	23	3.6
L	RdRp	7 (5)	3945	153 (32)	7 (0–60)	32	0.8
S, M, and L	All	-	6144	668 (391)	-	164	2.7

* Sites with at least two alleles. ^†^ Sites with at least two alleles, where each of those alleles is found in at least two sequences. ^‡^ Sites displaying amino acid residues unique to the genotype II clade. ^§^ Genotype II strains, excluding Mbarara strain.

## Data Availability

All consensus sequences generated and analyzed for this study were uploaded to NCBI GenBank under accession numbers MW452933–MW452935. Raw sequence data have been deposited in ENA under study PRJEB42607. Nucleotide and amino acid multiple sequence alignments and maximum likelihood phylogenies for each segment are available at: DOI: 10.5281/zenodo.4455021.

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
