# Peer review of "Phylogenetic Characterization of Crimean-Congo Hemorrhagic Fever Virus Detected in African Blue Ticks Feeding on Cattle in a Ugandan Abattoir"

_microorganisms, 2021, doi:10.3390/microorganisms9020438_

Round 1

Reviewer 1 Report

The work “Phylogenetic characterization of Crimean-Congo haemorrhagic fever virus detected in African blue ticks feeding on cattle in a Ugandan abattoir” is a conscientious and well presented work concerning the taxonomy of Crimean Congo Hemorrhagic Fever Virus (CCHFV). The new CCHFV virus (Mbarara) was isolated from ticks, fully sequenced, and its systematic position on the evolutionary trees of CCHFV segments was established. It was shown that the S and L segment are reliably located in the Genotype II cluster (Africa 2), while the M segment differs greatly from the M segments of the closest strains. The greatest similarity to M segments of Mbarara strain have several sequences from India and China, indicating that the M segment had likely been obtained through re-assortment of Asian and African viruses, through intercontinental movement of CCHFV-infected ticks on livestock or migratory birds. Work of Wampande and al. is one of the few works related CCHFV viruses isolated in nature. Meanwhile, knowledge of the spectrum of CCHFV variants in animal and vector reservoirs is needed to control the circulation of the virus and may prove useful for the design of future vaccines.

Author Response

We thank the reviewer for their comments.

Reviewer 2 Report

This manuscript describes the survey of ticks on livestock, looking for strains of the Crimean-Congo hemorrhagic fever virus in Uganda, a region poorly surveyed for the virus.   I modestly think the paper reports new and interesting data. I would like to see more samples, and not only 70 ticks, but in any case, the data is new and welcome. I do not have any major comments on the methodology and the conclusions, but I would like to kindly recommend the authors a few questions, mainly stylistic: - Please adhere to the international rules of zoological nomenclature: always use scientific names in italics, spell out the generic name when first time used and then later use only the initial (exception, after a period or at the beginning of a paragraph). - I think it is not necessary to remind that R. decoloratus is in the subgenus Boophilus. This is something known since the year 1999, and I think it is a bit outdated when continuously used in this paper. - I cannot catch the meaning of the Figure 4 in the context fo this manuscript: it is interesting, but in my humble opinion is out of context. If the authors want to keep this figure, I would kindly suggest to explain better “why” and then “milk” this data in the results and the discussion.  - I do not know if my next question is feasible: for me, providing the names of the countries (i.e. China) of isolation of a viral strain means nothing (China is a continent by itself) and do not reflects adequately the conditions and the ticks (or vertebrates, not cited in the paper) in which these strains were recorded. However, I know very well that many sequences in GenBank lack data about location, coordinates, etc. I would suggest the authors to include data about locality (and vertebrate or tick) of collection if available, that would enrich a lot this paper. A call of caution: sometimes, CCHF viruses are sequenced *after* passage from the tick to newborn mice by several generations; then, the virus tend to “loose” some portions of its segments (that reflect “adaptation to the tick”) and become more homogeneous (since adapted to the new laboratory host). I would suggest a caution when comparing similarities among strains or “conservatism” of parts of the RNA in different viral strains if they come from ticks (questing or feeding) or form vertebrates - I must to thank the authors for providing the support for each branch of the phylogenetic trees: this is something that is missing in many papers (this is not a question, it is just a sincere acknowledgment).    Thank you.

Author Response

We wish to thank the reviewer for their constructive comments. We agree that more samples (and more strains of CCHFV) would add to this study. As this was intended as a pilot study, we hope to provide such data in the future. With regard to the reviewer’s comments:

  1. The style has been amended for all Linnaean names and mentions of subgenus Boophilus have been removed from the text other than in the abstract and the first instance.
  2. Figure 4 was intended as a straightforward way for the reader to digest the variability over the M segment, highlighting the fact that this variability is not merely restricted to the hyper-variable mucin-like domain but is in also present in the various glycoprotein coding regions. We believe this is relevant as these glycoproteins are likely to form the basis of future vaccine designs. We have amended the text in the results and discussion to hopefully make this intention clearer.
  3. The reviewer’s suggestion to include location and host specifics is an excellent one and has been implemented in the text.
  4. We have also added a short note to the discussion regarding the implications of passaging the virus prior to sequencing (as is the case for the Chinese strains discussed).

Thank you.